# PERFORMANT LLM AGENTIC FRAMEWORK FOR CONVERSATIONAL AI

## ABSTRACT

The rise of Agentic applications and automation in the Voice AI industry has led to an increased reliance on Large Language Models (LLMs) to navigate graph-based logic workflows composed of nodes and edges. However, existing methods face challenges such as alignment errors in complex workflows and hallucinations caused by excessive context size. To address these limitations, we introduce the Performant Agentic Framework (PAF), a novel system that assists LLMs in selecting appropriate nodes and executing actions in order when traversing complex graphs. PAF combines LLM-based reasoning with a mathematically grounded vector scoring mechanism, achieving both higher accuracy and reduced latency. Our approach dynamically balances strict adherence to predefined paths with flexible node jumps to handle various user inputs efficiently. Experiments demonstrate that PAF significantly outperforms baseline methods, paving the way for scalable, real-time Conversational AI systems in complex business environments.

## 1 INTRODUCTION

Graph-based workflows are central to numerous business processes across industries such as education, legal, healthcare, and customer support. These workflows represent decision-making steps as nodes, and connections between them as edges. The rise of Conversational AI within these spaces introduces new challenges. Autonomous agents, powered by large language models (LLMs), are increasingly being used to navigate these workflows, enabling the automation of complex business processes (Zhuge et al., 2023). Each node in the workflow contains specific instructions or prompts that guide the agent's speech generation and certain actions to trigger. Nodes can be classified into several types, including Start Nodes, which define the root and entry point of a workflow; End Nodes, which signal the termination of the workflow; and generic Nodes, which serve as intermediate decision points containing speech instructions for the LLM to converse with users in predefined ways. Additionally, Transfer Nodes in Conversational AI workflows allow for transitioning the conversation to another autonomous or human agent. Edges between nodes may include logical conditions that dictate the agent's transitions, ensuring workflows are executed accurately.

Figure 1 illustrates how tasks such as determining health care eligibility can be broken down into nodes, edges, and conditions. For example, a healthcare provider might use such a workflow to efficiently filter out patients without the required insurance, reducing the burden on human agents. However, workflows like these can rapidly grow in complexity. As shown in Figure 2, adding just a few additional conditions to the conversation flow can drastically increase the number of nodes and edges, making the workflow more difficult to manage and execute effectively.

Although LLMs such as GPT and LLAMA are built on autoregressive decoder-based transformer architectures optimized for natural language generation, they are not inherently designed to handle structured, multi-step processes with extensive context (Qiu & Jin, 2024; Shi et al., 2023). Existing approaches have been to add a planning phase, where the LLM would take time to orchestrate the action, and then proceed to the generation tasks (Valmeekam et al., 2023; Zhou et al., 2024). However, this approach is not optimal to the Conversational AI use case, as it would increase the overall latency by doubling the number of queries needed. Tasks such as managing end-to-end customer service requests with non-standard return policies, performing outbound sales calls that involve dynamic CRM updates, or redirecting users to appropriate departments after a sequence of filtering questions require precision, alignment, and low-latency responses. These limitations force

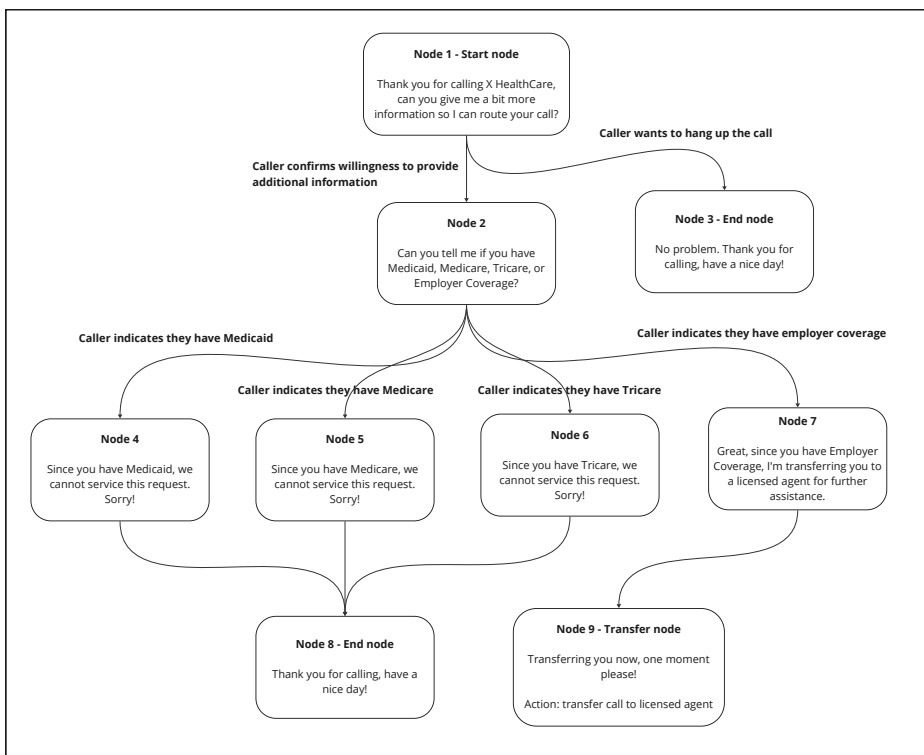

Figure 1: Example illustration of an Agentic workflow for a healthcare call center use case, where the Agent needs to route calls based on different conditions.

businesses to oversimplify workflows, sacrificing accuracy and operational efficiency—an outcome that is contrary to their objectives.

The challenges inherent in adapting LLMs to graph-based workflows underscore the need for new approaches that can accurately and efficiently execute workflows while respecting real-world constraints such as latency. While adding more reasoning steps could theoretically improve accuracy, such methods are impractical for Conversational AI applications where rapid response times are critical.

To address these challenges in the current Conversational AI space, this paper introduces the **Performant Agentic Framework (PAF)**, a novel solution for efficient graph traversal that balances accuracy and latency in real-world applications. By leveraging both traditional decision-making logic and mathematical methods for next-node selection, PAF enables agents to execute workflows with greater precision and speed. Our experiments demonstrate that PAF significantly outperforms baseline and traditional methods in both accuracy and latency, as evidenced by higher alignment scores and reduced response times.

## 2 RELATED WORK

The reliance on LLM-based systems to execute graph-based workflows has seen significant research attention, particularly in developing frameworks that aim to balance accuracy, latency, and alignment with predefined workflows. Below, we discuss prominent related works and their limitations.

**Agentic Frameworks.** Serving as examples, LangChain (LangChain, 2023) and LangGraph (LangGraph, 2023) streamline graph-based workflows by utilizing function calls and prompt chaining. While effective for simple tasks, their reliance on keyword-based triggers often results in alignment errors, especially in workflows with hundreds or thousands of nodes. These frameworks lack robustness for real-world applications where actions must be dynamically triggered at various points

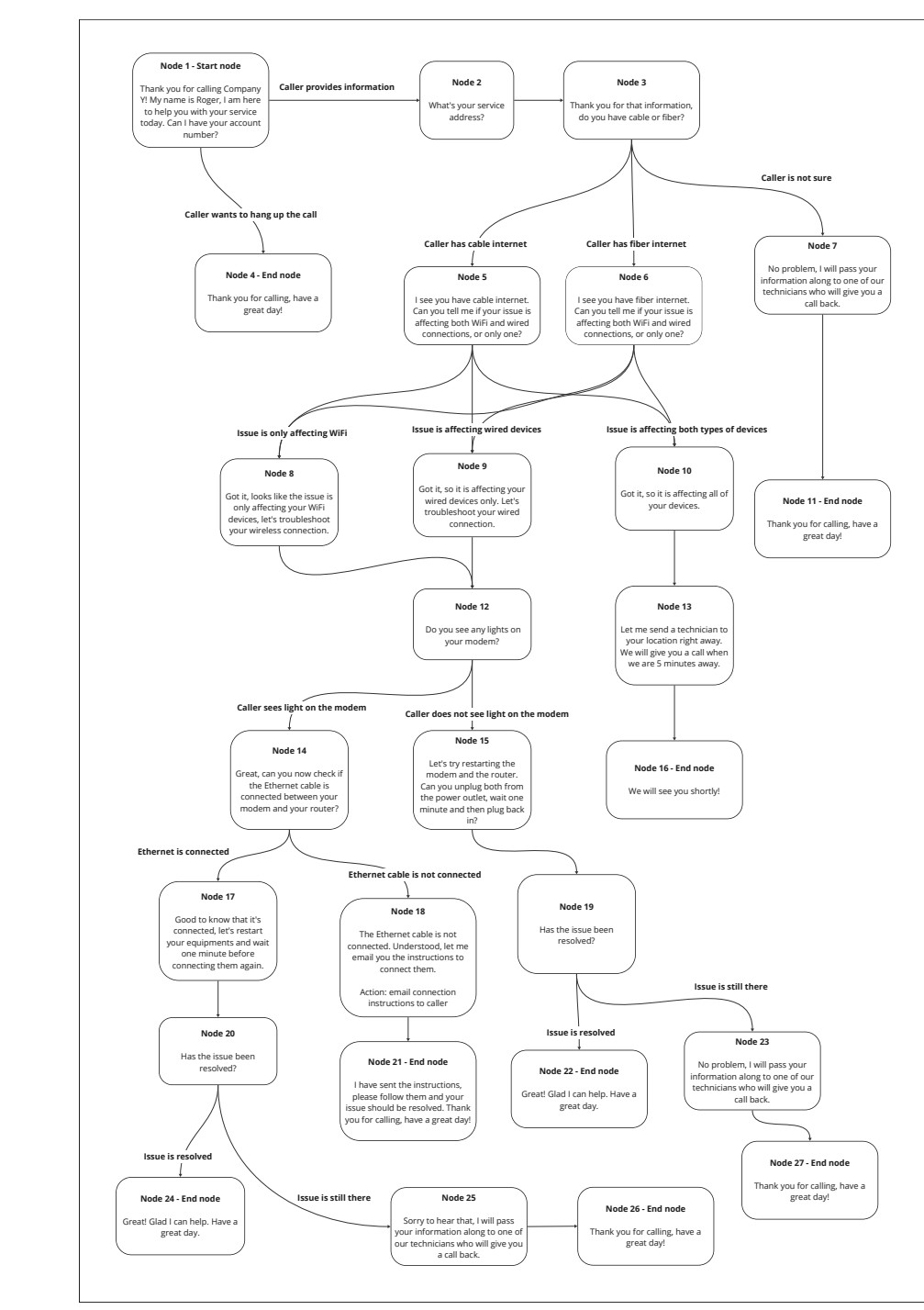

Figure 2: Example illustration of an Agentic workflow for an internet service company helping callers troubleshoot connection issues. This workflow demonstrates how a more complex use case can have more conditions, nodes, and edges.

in conversations. Furthermore, their reliance on LLM-generated triggers leads to unreliability in critical workflows, where adherence to predefined paths is essential for compliance and business logic (LangChain, 2023; LangGraph, 2023). Additionally, limitations in LLM context windows further

exacerbate their inefficiency in retaining relevant information across extended workflows, introducing hallucinations and context drift during execution (Dong & Qian, 2024).

**Conversational AI.** Conversational AI has been a key focus for Natural Language Processing. Existing Conversational AI solutions emphasize the need for multi-modality, guardrails, and advanced tuning to enhance dialogue quality (Dong et al., 2023). Prior approaches to the Voice AI space have proven to work in sandbox conversational settings (James et al., 2024), but lack the consistency and accuracy required for production use. As noted, LLMs often miss certain abilities to maintain performance in a dynamic conversational setting, unable to handle numerous tasks conditionally while reducing hallucinations and staying within context (Gill & Kaur, 2023; Dong et al., 2023; Dong & Qian, 2024).

**MetaGPT and SOP Translation.** MetaGPT leverages Standardized Operating Procedures (SOPs) to structure workflows, enabling agents to replicate domain-specific expertise. However, its reliance on iterative planning and validation increases latency, making it unsuitable for real-time applications. For example, Gao et al. (2023) note that the planning phase requires additional LLM calls, which adds computational overhead. While MetaGPT is effective for SOP alignment, it struggles with unusual user inputs and extended workflows, leading to significant context drift. Its dependence on domain-specific fine-tuning also hinders generalizability (Gao et al., 2023; Wang & Liu, 2024).

**Comparison and Our Contributions.** Existing frameworks have made valuable contributions but are hindered by issues such as context drift, high latency, and alignment errors. PAF addresses these limitations by replacing LLM planning phases with a mathematical decision-making approach, combining vector-based node selection and specialized prompt engineering. Unlike previous methods, PAF reduces context size while improving accuracy, making it a scalable and production-ready solution for navigating graph-based workflows.

# 3 PERFORMANT AGENTIC FRAMEWORK (PAF)

PAF is a framework designed for Agentic workflows, enabling agents to navigate graph-based structures composed of nodes and edges to execute predefined workflows. It is comprised of two components: Basic PAF and Optimized PAF, each tailored to address specific challenges in workflow execution.

## 3.1 BASIC PAF

**Problem Formulation.** PAF enables agents to operate by following nodes connected by logical edges. During each generation turn, the agent follows the nodes in the graph according to the logical conditions specified as outcomes of the node. If a condition is met, the agent navigates to and executes the instructions of the next node in the graph.

**LLM as a Judge for Node Identification.** We leverage an LLM to identify the Agent's location in the map dynamically at each generation:

This design is particularly effective in production AI systems as it separates the generation tasks from other downstream modules, like Text-to-Speech (TTS). This modular approach optimizes latency by enabling parallel processing by TTS or other services. Compared to implementations where prompts are added in a single body, Basic PAF achieves lower error rates by using a step-by-step logic tree and reducing the need for additional validation iterations (Li & Yuan, 2023; Reddy & Gupta, 2021).

## 3.2 OPTIMIZED PAF

While Basic PAF offers significant improvements, it faces bottlenecks when workflows have many nodes (e.g., 50 nodes with 4 conditions each). These bottlenecks arise as the agent struggles to differentiate between semantically similar prompts on different branches.

**Vector-Based Node Search.** Optimized PAF addresses this by adding a vector-based scoring mechanism to reduce the context window size and improve logical adherence:

**Optimized Agentic Framework.** Integrating the Vector-Based Node Search:

---

**Algorithm 1** LLM as a Judge for Node Identification

---

**Input:** $ConversationHistory$, $NavigationMap$, $LastestIdentifiedNode$
**Output:** $UpdatedLatestIdentifiedNode$
**Step 1: Format Input for the LLM**
 Construct a prompt using $ConversationHistory$.
 Add a contextual anchor by traversing from the StartNode to $LastestIdentifiedNode$ and collect the first-layer child Nodes in the map, e.g., `"You were previously on Node {LastestIdentifiedNode} with options..."`.
 If $LastestIdentifiedNode$ is unavailable, use: `"This is the start of the task {task}, proceed to Node 0."`
**Step 2: Query the LLM**
 Send the question: `"Based on your latest responses, where are you currently in the navigation map?"`
**Step 3: Process the Response**
 Parse the response to identify the node mentioned by the LLM.
 Validate the identified node against $NavigationMap$.
**Step 4: Return the Result**
 Output the validated node as $UpdatedLatestIdentifiedNode$.

---

**Algorithm 2** Basic Agentic Framework

---

**Input:** $ConversationHistory$, $NavigationMap$, $LatestIdentifiedNode$
**Output:** $UpdatedLatestIdentifiedNode$
**Step 1: Initialize LLM Instructional Message**
 Construct an instructional prompt for the LLM agent.
 Add $ConversationHistory$ to the prompt in a structured way.
 Include instructions based on $LatestIdentifiedNode$.
 Construct a navigation prompt by traversing $NavigationMap$ and collecting all first-layer children nodes' instructions from StartNode to $LatestIdentifiedNode$.
**Step 2: Query the LLM**
 Send the user question: `"Based on the navigation map and your current node, respond to the user question: {user question}."`
**Step 3: Process LLM Output in a Streaming Loop**
**while** LLM agent streams output **do**
 (a) Identify Current Node via Algorithm 1.
 (b) Update $LatestIdentifiedNode$ to the new identified node.
 (c) Trigger any actions related to $LatestIdentifiedNode$.
 (d) Update $NavigationMap$ if necessary.
**end while**
**Step 4: Return the Updated Node**
 $UpdatedLatestIdentifiedNode \leftarrow LatestIdentifiedNode$

---

Compared to cosine similarity, our experiments found that the dot product captures subtle differences in magnitude, beneficial for domain-specific jargon and context (Huang & Wang, 2021).

## 4 EXPERIMENT

We designed experiments to evaluate PAF against baseline methods in graph traversal and node selection. We focus on latency, accuracy, and alignment with business logic, critical for Conversational AI.

### 4.1 SETUP

**Dataset Generation.** We generated a synthetic dataset simulating real-world workflows, each entry consisting of:

- **SystemPrompt:** A node navigation map with Agentic logic.

---

**Algorithm 3** Vector-Based Node Search

---

**Input:** $NavigationMap, LatestIdentifiedNode, Threshold, LatestAgentResponse$
**Output:** $UpdatedLatestIdentifiedNode$
**Step 1: Vectorize Instructions & Agent Response**
    Compute embeddings for $LatestIdentifiedNode$, its child nodes, and $LatestAgentResponse$.
**Step 2: Compute Similarity Scores**
    Use dot product or another similarity metric between $LatestAgentResponse$ embedding and each node's embedding.
**Step 3: Select the Best-Matching Node**
    Choose the node with the highest similarity.
    If the top score $> Threshold$, return that node as $UpdatedLatestIdentifiedNode$.
**Step 4: Update or Fall Back**
**if** score above Threshold **then**
    return $UpdatedLatestIdentifiedNode$
**else**
    return `false` (fall back to LLM-as-Judge)
**end if**

---

**Algorithm 4** Optimized Agentic Framework

---

**Input:** $ConversationHistory, NavigationMap, LatestIdentifiedNode, Threshold$
**Output:** $UpdatedLatestIdentifiedNode$
**Step 1: Precompute Node Embeddings**
    Vectorize instructions for all nodes in $NavigationMap$.
**Step 2: Format Input for the LLM**
    Include $ConversationHistory$, instructions for $LatestIdentifiedNode$, plus first-layer children instructions.
**Step 3: Query the LLM Agent**
    Send the constructed message to the LLM agent.
**Step 4: Process LLM Output in a Streaming Loop**
**while** LLM outputs tokens **do**
    (a) Perform Vector-Based Node Search (Alg. 3).
    **if** a node is found with high confidence **then**
        Update $LatestIdentifiedNode$ accordingly.
    **else**
        Fallback to LLM-as-Judge (Alg. 1).
    **end if**
**end while**
**Step 5: Trigger Actions & Update Graph**
    Execute any node-specific actions and update $NavigationMap$ if needed.
**Step 6: Return Result**
    $UpdatedLatestIdentifiedNode \leftarrow LatestIdentifiedNode$

---

- **ConversationHistory:** Turn-by-turn interactions.
- **GoldenResponse:** A verified reference response for alignment and context accuracy.

Conversations ended either upon reaching a terminal node or after a random turn limit (6–10). Golden responses were manually verified to ensure correctness.

## 4.2 EVALUATION METRICS

- **Semantic Similarity:** Alignment (0–1) with the golden response using OpenAI's text-2-vec-3-small embeddings (OpenAI, 2025).
- **Total Complete Hit Rate:** Percentage of responses exceeding a similarity threshold (0.97).
- **Mean and Median Similarity Scores:** A measure of overall alignment.

**Methods Compared:**

1. **Baseline:** Naive single-shot approach with the entire conversation and map in one prompt.
2. **Basic PAF:** Step-by-step logic (Algorithms 1 and 2).
3. **Optimized PAF:** Vector-based approach (Algorithms 1, 3, and 4).

### 4.3 HYPOTHESES AND STATISTICAL ANALYSIS

**H1:** Basic PAF has higher mean similarity than Baseline. **H2:** Optimized PAF has higher mean similarity than Baseline. **H3:** Optimized PAF has higher mean similarity than Basic PAF.

We used one-sided paired t-tests (significance $\alpha = 0.05$).

### 4.4 RESULTS

Table 1: Result Metrics Across Algorithms

| Method | Total Hits | Count ¿ 0.8 | Mean | Median |
|---|---|---|---|---|
| Baseline | 0 | 0 | 0.391 | 0.387 |
| Basic PAF | 16 | 14 | 0.481 | 0.400 |
| Optimized PAF | 35 | 22 | 0.594 | 0.496 |

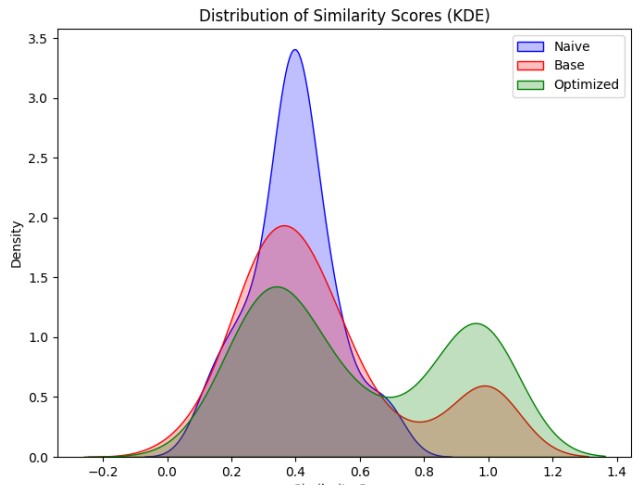

Figure 3: Distribution of Similarity Scores for the three tested frameworks. "Naive" is Baseline, "Base" is Basic PAF, and "Optimized" is Optimized PAF.

Table 2: Statistical Comparison Results (One-Sided Paired t-Tests)

| Comparison | t-statistic | p-value |
|---|---|---|
| Baseline vs Basic PAF | 2.9982 | 0.0020 |
| Baseline vs Optimized PAF | 7.3077 | 0.0000 |
| Basic PAF vs Optimized PAF | 4.2494 | 0.0000 |

**Findings.**

- **H1:** Basic PAF outperforms Baseline (p = 0.002).
- **H2:** Optimized PAF outperforms Baseline (p < 0.001).
- **H3:** Optimized PAF outperforms Basic PAF (p < 0.001).

## 4.5 REPRODUCIBILITY

We provide the code for data generation, evaluation, and visualization in an anonymized repository.[1]

## 5 CONCLUSION

Our approach introduces novel mechanisms for leveraging LLMs to navigate graph-based workflows, replacing the need for extensive planning phases and minimizing error rates. PAF achieves faster response times and greater accuracy in real-world applications by reducing reliance on large context windows and optimizing computational steps.

**Key contributions** include:

- Removing extra iterations for validation and planning, reducing latency.
- Improving alignment with a step-by-step logic tree that adds instructions incrementally, governed by a predefined logic.
- Reducing the context window size by focusing only on relevant graph information.
- Introducing vector-based scoring to reduce LLM calls and prioritize high-confidence next-node matches.

These improvements yield a more stable and controllable approach to help LLMs navigate workflows, trigger actions, align with business goals, and reduce hallucinations—all with improved performance in production settings.

## 6 FUTURE WORK

While Conversational AI serves as a compelling demonstration, PAF is broadly applicable. Planned extensions include:

- **Node Weights and Path Rules:** Allowing more flexible graph structure with weighted edges and dynamic transitions.
- **Integration with Other Models:** Evaluating domain-specific or smaller models to see how well they adapt to graph-based workflows.
- **Open-Source Model Improvements:** Adapting PAF to emerging open-source LLMs to enhance domain-specific tasks.

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
