# OpenReview forum: "Performant LLM Agentic Framework for Conversational AI"
_ICLR.cc/2025/Workshop/AgenticAI — ICLR 2025 Workshop AgenticAI Reject_

### Official Review · Reviewer_GMGB · 2025-03-03
**This paper introduces the Performant Agentic Framework (PAF), a novel approach designed to enhance the performance of Large Language Models in managing complex, node-based conversational workflows.**

**Rating:** 3
**Confidence:** 4

**Review:**

### **summary of strength**:
1) **Motivation**: The motivation behind PAF is clear and pragmatically grounded in real-world application needs, particularly within environments requiring dynamic conversational AI capabilities

2) **New framework**: The paper introduced new framework for node selection. Result show improvement in accuracy of the task.
### **summary of weakness**:
1) **Lack of comparison to other baselines**: While the paper acknowledges several existing frameworks such as LangChain and MetaGPT in the related work section, it notably lacks direct comparisons with these approaches, limiting the evaluation to a simplistic "naive single-shot" baseline. This restricts the ability to assess PAF's true advancements over potentially more sophisticated and similarly aimed approaches currently in use.

2) **Lack of description for experimental setting**: The experimental section of the paper fails to detail the specific settings used, including the backbone models, datasets, and configuration parameters.

3) **Lack of analysis on vector-based scoring**: The vector-based scoring mechanism is a core component of Optimized PAF, yet there is no ablation study isolating its contribution. It is unclear how much this mechanism enhances overall system performance.

4) **Contribution to the literature**: The paper claims that the framework can effectively address shortcomings in the literature, such as "context drift" and "alignment errors," yet it does not provide empirical evidence or specific experiments to substantiate these claims.

5) **typos in table 1**: The header of table 1, column2 is not clear.

### **Questions**

1) Can you provide an experimental comparison against state-of-the-art agentic frameworks such as LangChain or MetaGPT? This would give a clearer picture of PAF’s advantages over existing solutions. How does PAF perform relative to these systems in terms of accuracy, efficiency, and scalability?

2) The contribution of the vector-based scoring mechanism to the overall performance improvement remains unclear. Could you conduct an ablation study where this component is removed or replaced with alternative selection techniques?

3) Given the role of the threshold in the vector-based node selection process, how was this threshold determined? Was it chosen heuristically, or was it tuned through systematic experimentation?

---

### Official Review · Reviewer_aynr · 2025-03-04
**Review for Performant LLM Agentic Framework for Conversational AI**

**Rating:** 6
**Confidence:** 3

**Review:**

Summary:
This paper proposes the Performant Agentic Framework (PAF) to improve LLM-driven Conversational AI in graph-based workflows. Existing methods struggle with alignment errors, hallucinations, and high latency, limiting real-world deployment. PAF addresses these issues by combining LLM-based reasoning with a vector-scoring mechanism for efficient node selection. It includes Basic PAF, which follows step-by-step logic trees, and Optimized PAF, which reduces context size using vector-based node selection. Experiments show that PAF significantly improves accuracy and response time, making it a scalable solution for real-time business applications.

Pros:
1.	The paper addresses specific challenges in navigating graph-based workflows with LLMs (high latency, alignment errors, hallucinations) that are genuinely problematic in production environments.
2.	The PAF framework combines LLM reasoning with vector scoring mechanisms, reducing latency while maintaining accuracy. This hybrid approach effectively addresses the efficiency issues of relying solely on LLM planning.

Cons:
1.	While the paper mentions frameworks like LangChain and LangGraph, it doesn't directly compare PAF against these established solutions, opting instead for custom baseline methods.
2.	The experiments are primarily based on synthetic datasets, lacking evaluation in real production environments, which limits the generalizability of the research conclusions.
3.	The paper lacks detailed analysis of computational resource requirements (memory, CPU/GPU demands) and doesn't explore how the framework's performance scales with workflows of different sizes.

---

### Official Review · Reviewer_eaKp · 2025-03-05
**Review for Performant LLM**

**Rating:** 4
**Confidence:** 4

**Review:**

This paper introduces the Performant Agentic Framework (PAF), a system designed to help Large Language Models (LLMs) navigate complex graph-based workflows in Conversational AI applications. Many existing AI-driven voice assistants struggle with accuracy and efficiency when following structured workflows, leading to errors in decision-making and slow responses. PAF improves this by using vector-based node selection combined with step-by-step logic trees, allowing AI to choose the right actions quickly and accurately while reducing the need for extra planning steps. Experimental results show that PAF outperforms existing methods in both accuracy and speed, making it a promising solution for scalable, real-time AI conversations in industries like customer service, healthcare, and tech support.

Strengths of the Paper:
1. Improves AI decision-making in complex workflows, reducing errors in automated conversations.
2. Uses structured logic trees and vector scoring, leading to better accuracy in AI responses.

Weaknesses of the Paper:
1. Limited to graph-based workflows, making it less useful for unstructured conversations.
2. Depends on predefined paths, which may reduce AI flexibility in handling unexpected scenarios.
3. Requires specialized setup and tuning, making it harder to integrate into new AI systems.

---

### Decision · Program_Chairs · 2025-03-05

Reject